# Threat Appraisal, Resilience, and Health Behaviors in Recovered COVID-19 Patients: The Serial Mediation of Coping and Meaning-Making

**DOI:** 10.3390/ijerph20043649

**Published:** 2023-02-18

**Authors:** Dariusz Krok, Ewa Telka, Małgorzata Szcześniak, Adam Falewicz

**Affiliations:** 1Institute of Psychology, University of Opole, 45-052 Opole, Poland; 2Department of Radiotherapy, Maria Sklodowska-Curie National Research Institute of Oncology, Gliwice Branch, 44-101 Gliwice, Poland; 3Institute of Psychology, University of Szczecin, 71-017 Szczecin, Poland

**Keywords:** threat appraisal, coping strategies, resilience, health behaviors, meaning-making, recovered COVID-19 patients

## Abstract

Research indicates that both cognitive appraisal and personal resources can noticeably influence health behaviors, as individuals modify their health convictions and practices on the basis of threat appraisal, personality, and meaning. The aim of the current study was to investigate whether coping strategies and meaning-making can serially mediate the relationship of threat appraisal and resilience with health behaviors in recovered COVID-19 patients. Self-report measures of threat appraisal, resilience, coping, meaning-making, and health behaviors were completed by 266 participants (aged 17 to 78, 51.5% female) who had recovered from COVID-19. The serial mediation analysis showed that the relationship of threat appraisal and resilience with health behaviors was mediated by problem-focused coping, meaning-focused coping, and meaning-making, but not by emotion-focused coping. These results suggest that associations among threat perception, resilience, and health behavior depend to some extent on the interplay of coping and meaning-making, which reveals their unique role in the process of recovery from COVID-19, with potential implications for health interventions.

## 1. Introduction

The outbreak of the COVID-19 pandemic, caused by the Severe Acute Respiratory Syndrome Coronavirus-2, has led to 6,833,388 deaths and affected more than 755,385,709 people worldwide, as of February 2023 [1]. According to statistics released by the Polish Ministry of Health, within the same period, 118,779 patients died in Poland due to COVID-19, and more than 6,389,572 recovered from this disease [2].

In many ways, the pandemic has left most people confused and unprepared to deal with the unpredictability of the disease and its consequences [3]. Moreover, there is substantial evidence that the coronavirus has had significant lasting and emergent side effects in recovered patients [4,5]. Systematic psychological reviews and meta-analyses of current studies show that COVID-19, besides having economic (e.g., lockdown, loss of work) and social consequences (e.g., quarantine, social distancing, isolation, loneliness), has had detrimental ramifications on mental health [6]. Infected people display higher levels of emotional difficulties [7], anxiety or depression [8,9,10], post-traumatic stress symptoms [10], sleep disturbances [9], chronic fatigue, and pain [11].

Still, little research has investigated the functional coping strategies and meaning-making processes that modified both negative and positive reactions to the pandemic [12,13]. Therefore, the aim of this study was to examine how threat appraisal and resilience relate to health behaviors in recovered COVID-19 patients in terms of coping and meaning-making. The identification of pathways (coping strategies and comprehension of challenging life events) through which assessment of the COVID-19 pandemic affects the health practices of recovered patients may contribute to further understanding the indirect relationships between disease-related appraisal and health behaviors.

### 1.1. Threat Appraisal, Resilience, and Health Behaviors

Health behavior is the fluctuating response to stressful events [12], and depends on several factors [14]. Research shows that people differ in their sensitivity to threats and resilience, assigning various meanings to life challenges [15].

Threat appraisal, which is defined as the subjective interpretation of the likelihood, acuteness, and vulnerability regarding a difficult situation [16,17], motivates people to action [18]. Some research conducted during the COVID-19 outbreak confirmed that threat appraisal is related to the implementation of preventive behaviors [19,20]. People who display increased threat and coping appraisal are likely to perceive health distress as manageable [21], employ effective strategies, and generate adaptive solutions [22]. Resilience, conceptualized as “a dynamic process wherein individuals display positive adaptation despite [or after] experiences of significant adversity or trauma” ([23], p. 858), can play a similar role. Recent evidence demonstrated that people who present higher levels of resilience are likely to show a lower degree of trait catastrophizing [24] and psychological distress [25], higher hope, optimism, happiness, life satisfaction [26,27], and better mental health [25] and emotional adjustment [28]. They are also more prompt in recovering from difficult situations to recuperate psychological balance. Thus, based on prior literature and assorted studies, we assumed the following:

**H1.** 
*Threat appraisal regarding a difficult situation, and a resilient ability to ‘bounce back’ from adversity [29], positively correlate with those behaviors that promote well-being, such as proper eating habits, preventive behaviors, health practices, and positive mental attitude.*


### 1.2. Coping Strategies and Other Variables

There is an internal connection between coping strategies and other variables considered in the current study. Coping, perceived as a conscious response to stress, is one potential reaction to threat [30] that affects physical and psychological health [31]. People who face stressful events attempt to manage or reduce the negative experience by taking direct action, behaving emotionally, or giving meaning to what is difficult [30]. Problem-, emotion-, and meaning-coping strategies are positively connected to psychological well-being [32]. A series of meta-analyses [33] demonstrated that problem-focused coping is positively associated with health, indicating that efforts to solve the problem characterize individuals who report adaptive health behaviors. It has been found that active forms of coping have a positive effect on people’s health, whereas avoidance has the opposite effect [34,35]. Likewise, highly resilient people tend to engage in active solutions [36], and largely choose problem-focused coping [37]. Stronger resilience was related to lower levels of anxiety during the COVID-19 pandemic [38]. Emotional coping is often considered less adaptive [39]. However, it seems that this form of strategy, depending on the circumstances, can also play a functional role. Indeed, it has been confirmed that emotion-focused strategies are associated with more growth [12]. Adaptive and functional emotional coping (e.g., emotional support) enables people to deal with stress correctly. Conversely, non-adaptive emotional coping (e.g., rumination) results in dysfunctional responses to difficult events [40]. Meaning-focused coping is an important regulation strategy during particularly stressful events [41], which helps to re-establish personal resources over a relatively long time as a result of meaning reappraisal [42]. It also predicts better adjustment, reflecting psychological resilience [43]. Hence, we hypothesized the following:

**H2.** 
*Problem-, emotion-, and meaning-focused coping correlate positively with threat appraisal, resilience, and healthy behavior.*


### 1.3. Coping Strategies and Meaning-Making as Mediators

While the direct relationship between threat appraisal/resilience and healthy behavior is empirically quite well documented, knowledge about the mechanisms behind this association is far less clear. Since sustaining health behavior can be difficult for recovered COVID-19 patients due to psychological distress and physical consequences [44], it seems crucial to identify patients’ resources which may play a protective role in maintaining well-being. Therefore, considering that the analysis of the mediation effects is crucial for explaining the outcomes [45], we investigated whether coping strategies and meaning-making could act as potential intermediary variables between threat appraisal/resilience and healthy behavior.

The main rationale for this choice was that both coping and meaning-making could serve as protective factors against unhealthy behaviors. Indeed, coping strategies have been found to be mediators between antecedent difficult situations and outcomes, such as different dimensions of health [30], or between personality and adjustment [46]. However, specific coping strategies can have different mediating functions. In a sample of healthcare personnel, problem- and meaning-focused coping, but not emotion-focused coping, mediated the relationship between risk perception of COVID-19 and psychological well-being, and between meaning-based resources and psychological well-being [32]. Likewise, meaning-making has recently been identified as a mediator between perception of COVID-19 and benevolent orientation toward others [47].

The mediational relationship between threat appraisal/resilience, coping strategies, meaning-making, and healthy behavior can be better grasped on the theoretical grounds of protection motivation theory (PMT) [48]. When people face fears related to a difficult situation or event (e.g., COVID-19), they act based on two cognitive processes: threat identification and evaluation, and the ability to cope with it. Depending on how individuals assess a threat, how resilient they are in facing it, what coping strategies they adopt, and what meaning they give to difficult events, they implement adaptive or maladaptive behavioral responses. This means that healthy behaviors are not only affected by threat appraisal/resilience, but mediating factors may also influence the association between these variables. Empirical research shows that coping with stressful events is regarded as the intermediate process that can alter the relationship between a stressful situation and health [45]. The alternating function of coping strategies may consist of facilitating or impeding mental and physical health [30]. For example, emotion-oriented and task-oriented coping acted as mediators between temperamental traits and occupational burnout [49], different types of perfectionism and perceived stress [50], resilience and post-traumatic stress disorder [51], and the motivational states of engagement and burnout [52]. Moreover, meaning-making mediated the relationship between self-efficacy and life satisfaction among healthcare workers during the first COVID-19 lockdown [48]. Likewise, the presence of meaning mediated the association between social support and post-traumatic growth [53]. Therefore, it can be concluded that identification of a threat and the ability to deal with it can enable people to give it appropriate meaning and adopt healthy behaviors. Based on theoretical considerations and empirical findings, we assumed the following:

**H3.** 
*Coping strategies and meaning-making are serial mediators in the relationship between threat appraisal/resilience and healthy behaviors.*


## 2. Materials and Methods

### 2.1. Participants and Procedure

Eligible participants were people who had recovered from COVID-19. We set the following inclusion criteria: (1) confirmed diagnosis of COVID-19 during illness; (2) cognitive abilities to complete the questionnaire; and (3) recovery from the symptoms of COVID-19. The exclusion criteria were the following: (1) serious medical conditions that would distort the person’s responses (cancer, cardiovascular disease, obstructive pulmonary diseases, diabetes); and (2) lack of ability to fill in the questionnaire due to cognitive impairment, disability, or severe distress. COVID-19 was confirmed by two main types of viral tests: nucleic acid amplification tests (NAATs) and antigen tests. Our study was cross-sectional in design.

### 2.2. Procedure

A purposive sampling method was used to select participants who were COVID-19 survivors. The study was conducted from 1 November 2021 to 31 March 2022 in southern parts of Poland (Opole, Wroclaw, and Katowice), both privately by individuals in their homes, and in outpatient clinics. Initially, 325 people were approached for the study: 31 declined to participate, and 28 turned out to be ineligible due to the inclusion vs. exclusion criteria. Finally, 266 people agreed to participate (the participation ratio was 81.84%). The participants were approached by study assistants who informed them about the study and handed them the questionnaire and written informed consent. The questionnaires had to be filled in at home and given back to the assistant. Debriefing was provided after the study, along with answers to potential questions. The University Ethics Board approved the research material and procedure.

### 2.3. Measures

The following questionnaires were used to assess five outcomes: threat appraisal, resilience, health behaviors, coping, and meaning-making.

Threat appraisal was assessed using the Perceived Threat of COVID-19 Scale [32], a six-item instrument using ratings from 1 (strongly disagree) to 5 (strongly agree). The scale evaluates the perceived threat severity of COVID-19 that reflects the person’s negative personal, societal, and economic consequences related to the pandemic. The Cronbach’s reliability for the present study was 0.82.

Resilience was measured using the Brief Resilience Scale (BRS), which is a reliable tool to quantify respondents’ resilience, understood as one’s ability to recover from stress [46]. The scale contains six items which are evaluated on a five-point Likert scale, ranging from 1 (strongly disagree) to 5 (strongly agree); higher scores indicate higher resilience. The Cronbach’s reliability for the present study was 0.84.

Health behaviors were measured with the Health Behavior Inventory [54]. This is a 24-item scale that measures individuals’ health practices which are conducive to health. Each item is answered on a five-point Likert scale, ranging from 1 (almost never) to 5 (almost always). The instrument measures four categories of pro-health behaviors: proper eating habits, including the type of food primarily consumed; preventive behaviors, referring to following health recommendations and seeking knowledge and information about health and disease; health practices, including daily habits related to sleep, recreation, or exercise; and positive mental attitude, comprising psychological factors in health behavior such as avoidance of high stress, emotions, tension, and depressing situations. The sum of the scores from all four scales indicates the overall health behavior score. Higher scores represent healthier behaviors. The inventory has demonstrated good reliability and validity in clinical and non-clinical groups [54]. The Cronbach’s coefficients for the present study were 0.85 for the total result, and from 0.75 to 0.86 for the subscales.

Coping was assessed using the Coping Questionnaire [55], which measures three coping strategies: problem-focused coping, emotion-focused coping, and meaning-focused coping. The questionnaire comprises 37 items scored on a five-point Likert scale, ranging from 1 (not at all) to 5 (very much). Higher scores represent greater use of the coping strategies. The Cronbach’s coefficients for the present study were 0.83 (problem-focused coping), 0.84 (emotion-focused coping), and 0.86 (meaning-focused coping).

Meaning-making was measured using the Meaning-Making Questionnaire [56], which assesses one’s cognitive ability to comprehend and integrate challenging life events into a coherent set of personal meaning, beliefs, and goals. It contains six items scored on a five-point Likert scale, ranging from 1 (never) to 5 (very often). Higher scores reflect a stronger and more dynamic process of meaning-making. The Cronbach’s reliability for the present study was 0.86.

### 2.4. Data Analysis

Statistical analyses were conducted using Statistica, version 13, and Amos, version 21. A priori power analysis G* showed that a sample size of 250 participants is sufficient (f = 0.10, α = 0.05) to obtain a required 90 percent power in our analysis [57]. Firstly, as part of the preliminary analyses, multicollinearity through VIF and tolerance were checked: the value of the variance inflation factor (VIF) collinearity test^2^ was 1.43, while the value of the tolerance coefficient was 0.67, which indicate an acceptable degree of correlation of the predictors entered for the analysis conducted. Secondly, bivariate correlation analyses were calculated among all variables. Thirdly, we decided to test the serial mediation effects using SEM analysis. The covariance matrix and maximum likelihood estimation were applied, and model fit was evaluated. Thirdly, the bootstrapping procedure (95% bias-corrected confidence intervals, 5000 bootstrap resamples) was applied to test direct and indirect effects between the variables [58]. The procedure examined the serial mediation effects of coping and meaning-making on the relationship between threat appraisal, resilience, and health behaviors. Missing data were controlled by the case-wise mean substitution method that calculates average score from the matching subscale. All models were adjusted for age, gender, and education.

## 3. Results

### 3.1. Sample Characteristics

The participants were aged between 18 and 78 years (M = 40.31; SD = 17.19). More clinical and demographic characteristics are provided in Table 1.

### 3.2. Descriptive Statistics and Correlations

The correlational results showed that the time since COVID-19 was positively associated with symptom severity, resilience, problem-focused coping, preventive behaviors, and health practices (Table 2). Age was positively correlated with all variables, except for threat appraisal. Symptom severity was positively correlated with threat appraisal, resilience, problem-focused coping, emotion-focused coping, meaning-focused coping, proper eating habits, preventive behaviors, health practices, and positive mental attitude. Threat appraisal was positively associated with problem-focused coping, emotion-focused coping, meaning-focused coping, and preventive behaviors. Resilience was positively related with problem-focused coping, emotion-focused coping, meaning-focused coping, meaning-making, proper eating habits, preventive behaviors, health practices, and positive mental attitude. Problem-, emotion-, and meaning-focused coping, as well as meaning-making, were positively associated with proper eating habits, preventive behaviors, health practices, and positive mental attitude.

### 3.3. Serial Mediation Effects

The confirmatory factor analysis of the measurement model demonstrated that the model had a very satisfactory fit to the data: *χ*^2^ (N = 266) = 104.21; *p* < 0.001; NFI = 0.95; CFI = 0.96; RMSEA = 0.03; SRMR = 0.03. All the factor loadings for the indicators (proper eating habits, preventive behaviors, health practices, and positive mental attitude) on the latent factor (health behaviors) were significant (*p* < 0.001).

Next, our research model was tested, including directional paths between threat appraisal, resilience, problem-, emotion-, and meaning-focused coping, meaning-making, and health behaviors. However, the initial model had a rather poor fit to the data: *χ*^2^ (29) = 143.39; *p* < 0.001; GFI = 0.85; AGFI = 0.79; CFI = 0.43; RMSEA = 0.13 [0.10; 0.14]; Hoelter’s index = 92. Additionally, some paths were statistically non-significant (e.g., the paths from threat appraisal to health behaviors, problem-, and meaning-focused coping, or the path from emotion-focused coping to meaning-making; *p* > 0.05). Thus, we decided to re-test that model to improve its fit, in accordance with the recommendations of Mueller and Hancock [59]. This resulted in a final model that showed a substantial improvement and satisfactory fit to the empirical data: χ^2^ (23) = 55.37; *p* < 0.001, GFI = 0.95; AGFI = 0.91; CFI = 0.88; RMSEA = 0.07 [0.05; 0.10]; SRMR = 0.05; Hoelter’s index = 201. The final model is presented in Figure 1. Comparison of the initial model with the final one showed a better fit of the latter: Δ*χ*^2^ (6) = 88.02, *p* < 0.001.

Finally, a bootstrapping procedure was used [60] to assess mediation effects (Table 3). Only one direct effect turned out to be statistically significant, that of resilience on health behaviors. In addition, four indirect effects were significant: threat appraisal on health behaviors through problem-focused coping, meaning-focused coping, and meaning-making; resilience on health behaviors through problem-focused coping, meaning-focused coping, and meaning-making, both of which indicate serial mediation; problem-focused coping on health behaviors through meaning-making; and meaning-focused coping on health behaviors through meaning-making. Thus, problem-focused coping, meaning-focused coping, and meaning-making were serial mediators. Finally, we conducted a multi-group analysis to check how the overall model was adjusted for age, gender, and education. The path coefficients for problem-focused coping, meaning-focused coping, and meaning-making as serial mediators proved to be insignificant across age (χ^2^ = 1.01, *p* > 0.05), gender (χ^2^ = 1.23, *p* > 0.05), and education (χ^2^ = 0.94, *p* > 0.05).

## 4. Discussion

The first purpose of the study was to examine relationships among threat appraisal, resilience, coping strategies, meaning-making, and health behaviors. The second aim was to verify serial mediation roles of coping strategies and meaning-making in the relationship of threat appraisal and resilience with health behaviors. Our findings mostly confirm the hypothesized associations. They also extend previous research on the application of PMT with the results of people who have already experienced COVID-19 infections [61]. To our knowledge, this is the first study to reveal indirect associations among threat appraisal, resilience, and health behaviors that occurred through coping and meaning-making in recovered COVID-19 patients.

The positive correlations that were found between threat appraisal and health behaviors (H1) are in line with PMT [62,63]. People who assess a situation to be threatening intensify behaviors that serve health. Other research has also shown that people who are confronted with a risk of being infected with COVID-19 tend to engage in behaviors that are protective against that risk, such as travel avoidance [64] or reduced use of public transport [65]. This shows that through the assessment of perceived severity and vulnerability, and taking into account the benefits of modified behavior, people decide to undertake behavioral changes by implementing a healthier mental and physical lifestyle.

Positive associations between psychological resilience and all of the dimensions of health behaviors (H1) corroborate results obtained in prior studies, and confirm a role of resilience as a personal resource in dealing with adversity. This finding aligns with the outcomes of a study in a multiple-phase-change single-case experimental design run by Jeste et al. [66], showing that even remotely administered resilience-focused behavioral intervention during the COVID-19 pandemic led to a reduction in perceived stress and an increase in happiness in a senior group. In other disease contexts, patients with high resilience were more likely to understand their adverse situation [67], achieve treatment response [68], cope with the ongoing demands of their illness [69], and effectively adapt to challenging conditions [70]. Such optimal adaptations, despite or after disease, are likely to occur because resilient patients are characterized by a strong and integrated sense of personal identity [71]. Moreover, resilience helps patients overcome distress and make sense of complex adaptation processes [72].

Problem-, emotion-, and meaning-focused coping correlate positively with threat appraisal and healthy behavior (H2) [73]. Empirical evidence shows that threat appraisal and coping are related in different domains. Elkayal et al. [74] showed that psychological distress is positively connected to problem- and emotion-focused coping. It has been also observed that problem-focused coping [75], emotion-focused coping [12], and meaning [76] are associated with less distress. Moreover, problem- and emotion-focused coping correlated positively with threat appraisal and more post-traumatic growth among university students in their psychological response to COVID-19 [77]. A positive association of problem-focused coping with threat appraisal and healthy behavior was also obtained by Wu et al. [64]. Their research showed that COVID-19-specific protective behaviors, which specifically address the infection risks, were positively connected with perceived threat avoidability of COVID-19, and led to reduced psychic anxiety. In a longitudinal study by Szczuka et al. [77], perceived COVID-19 illness severity (for the higher general anxiety group) at T1 predicted higher perceived illness severity at T3, which led to higher intentional handwashing and handwashing adherence at T3. Our study also demonstrated that meaning-focused coping positively correlated with behaviors that promote well-being. This result is in line with those of previous studies, in which meaning-focused coping was positively associated with positive affect (including pain control) over time in somatic patients [55], and meaningful attitudes to transcendence were associated with a lower level of psychological distress during the COVID-19 pandemic [78]. Similarly, Ziółkowska et al. [79] showed that meaning-focused coping (measured as positive reappraisal) was a predictor of positive emotions in children diagnosed with cancer.

The positive correlations found between resilience and problem-, emotion-, and meaning-focused coping (H2) are also in line with those of previous studies which showed that individuals with higher levels of resilience tend to deal effectively with life problems [80,81], regulate emotions [80], and give meaning to stressful situations. Therefore, resilience enables individuals to use several protective strategies [82], assuming active coping, self-regulation, and making sense of life events [52]. Our findings are understandable if we take into account that, in the face of distress, resilient people possess a wide range of resources to deal with difficult events.

In difficult times, such as the pandemic and recovery from the disease, resilience can be optimized by finding meaning in life [83]. Meaning-centered activities were also associated with problem- and emotion-focused strategies [84], which may suggest that people who give meaning to difficult experiences are more inclined to adopt task- and emotion-centered coping. Finally, Eisenbeck et al. [84] confirmed that meaning-centered coping was a significant predictor of psychological and physical health during the COVID-19 pandemic.

Both coping strategies and meaning-making were found to be mediators in the relationship between threat appraisal and healthy behaviors (H3), and between resilience and healthy behaviors. These findings are very interesting, as they suggest that when people who recovered from COVID-19 assess the risk of the disease or its consequences with a resilient ability, they are able to adopt active coping, effectively deal with hardships and uncertainty, and modify the problem. They also make sense of the disease, since the restoration of the meaning hidden in every disruptive condition is the main goal of coping [85]. Our findings extend previous research on meaning in life and quality of life [86] by demonstrating that meaning-centered activities (i.e., meaning-focused coping and meaning-making) play a significant role in shaping healthy behaviors that are part of overall quality of life. Meaning-making helps reframe negative experiences and make sense of them [87], leading people to undertake healthy behaviors. In addition, threat appraisal and resilience necessitate the use of coping strategies which facilitate cognitive processes, and help identify positive meanings [88] for healthy behaviors.

## 5. Limitations

Although our study provides a better understanding of the relationships among threat appraisal, resilience, and healthy behaviors within the mediating effect of coping strategies and meaning-making processes, it also has some limitations. Firstly, the most important weakness is the lack of random group selection and inclusion criteria that were too general. In the case of purposive selection, we may have missed people who experienced the disease in a more severe way. Therefore, future studies should include more detailed data on the severity of COVID-19 (e.g., presence of life-threatening course of the disease, presence of comorbidities, use of oxygen therapy, etc.), which could prove to be moderating factors of the aforementioned relationships. Another important limitation of our study is its cross-sectional nature, which prevented us from deducing any causal associations among all the variables considered. Mediating relationships and their directionality, although supported by sound theory, are no substitute for a longitudinal study. A prospective study could examine the stability of the studied traits over time, while controlling for their baseline levels. In future studies, it would be constructive to undertake an experimental approach to confirm causal relationships and the role of variables. Another limitation is that only those who had contracted COVID-19 were studied, with no reference group. Future research could test the existence of the described relationships in people who did not become ill, since the imagined possibility of passing on the disease could activate similar coping resources, and through them, appropriate health-oriented behaviors. Another interesting direction for future research would be to examine the existential resources used by the respondents (e.g., philosophy of life, religiousness, values) in the process of meaning-making. Moreover, we did not consider any confounder variables. Given that the research group was diverse in terms of age, it can be assumed that age-related variables, such as general health, economic conditions, and satisfaction with interpersonal relationships, could have affected the overall mediation results.

## 6. Conclusions

Thus far, the present study is one of few studies that has explored the relationships among threat appraisal, resilience, and health behaviors within a mediational role of coping strategies and meaning-making. Two coping strategies—problem- and meaning-focused coping and meaning-making—were serial mediators in the relationship between threat appraisal and health behaviors. Our findings also showed that problem- and meaning-focused coping and meaning-making were serial mediators in the relationship between resilience and health behaviors, which, despite the threats related to COVID-19, provide opportunities to develop healthy behaviors through meaning-making processes. Finally, the study showed that in the process of treatment and therapy, as well as in the promotion of health, it is essential to emphasize the promotion of personal resources (e.g., religiosity, attentiveness, insight), which can enhance meaning-making processes.

## Figures and Tables

**Figure 1 ijerph-20-03649-f001:**
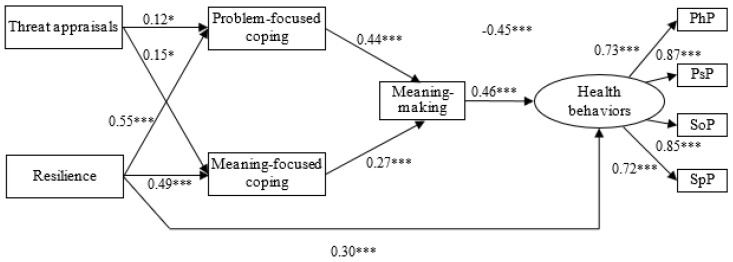
Final model of the relationships between threat appraisal, resilience, and health behaviors: serial mediation of coping and meaning-making (standardized coefficients). * *p* < 0.05; *** *p* < 0.001.

**Table 1 ijerph-20-03649-t001:** Clinical and demographic characteristics of recovered COVID-19 patients (N = 266).

Characteristics		
Age	M 40.31	SD 17.19
Body mass index	26.64	3.21
Symptom severity (on a scale from 1 to 10)	5.78	2.34
Time since COVID-19 in months	6.79	4.29
Range of the time since COVID-19	from 5 days to 18 moths	
	N	%
Sex		
Male	129	48.5
Female	137	51.5
Hospitalization due to COVID-19 symptoms	78	29.3
Educational attainment		
Basic education	40	15.1
Vocational education	75	28.3
High school education	101	37.9
University education	50	18.7

**Table 2 ijerph-20-03649-t002:** Correlations among medical factors, threat appraisal, resilience, coping, meaning-making, and health behaviors in recovered COVID-19 patients.

Variables	1	2	3	4	5	6	7	8	9	10	11	12	13
1. Age	-												
2. Time since COVID-19	0.36 ***	-											
3. Symptom severity	0.59 ***	0.29 ***	-										
4. Threat appraisal	0.05	0.10	0.20 **	-									
5. Resilience	0.46 ***	0.25 ***	0.30 ***	0.03	-								
6. Problem-focused coping	0.30 ***	0.13 *	0.27 ***	0.17 **	0.53 ***	-							
7. Emotion-focused coping	0.20 **	0.09	0.14 *	0.19 **	0.40 ***	0.63 ***	-						
8. Meaning-focused coping	0.16 **	0.13	0.18 **	0.17 **	0.49 ***	0.73 ***	0.73 ***	-					
9. Meaning-making	0.17 **	0.12	0.04	0.07	0.46 ***	0.59 ***	0.43 ***	0.56 ***	-				
10. Proper eating habits	0.25 **	0.07	0.16 **	0.09	0.19 **	0.28 ***	0.36 ***	0.29 ***	0.24 ***	-			
11. Preventive behaviors	0.35 ***	0.16 *	0.36 ***	0.21 ***	0.32 ***	0.37 ***	0.34 ***	0.36 ***	0.40 ***	0.59 ***	-		
12. Health practices	0.17 **	0.18 **	0.18 **	0.06	0.48 ***	0.42 ***	0.41 ***	0.41 ***	0.43 ***	0.47 ***	0.59 ***	-	
13. Positive mental attitude	0.11	0.11	0.22 ***	0.09	0.33 ***	0.39 ***	0.36 ***	0.38 ***	0.39 ***	0.46 ***	0.61 ***	0.54 ***	
M	40.15	6.80	5.79	4.20	3.44	3.86	3.77	3.85	3.70	3.47	3.64	3.67	-
SD	17.05	4.31	2.34	0.66	0.98	0.80	0.80	0.79	0.89	0.92	0.78	0.70	3.53
													0.62

* *p* < 0.05; ** *p* < 0.01; *** *p* < 0.001.

**Table 3 ijerph-20-03649-t003:** Bootstrapping indirect effects (standardized) and their 95% confidence intervals.

Model Pathways	Estimates	Standard Errors	95% CI	
			Lower	Upper
Threat appraisal→Problem- and Meaning-focused coping/Meaning-making→Health behaviors	0.04 ^a^	0.02	0.01	0.07
Resilience→Problem- and Meaning-focused coping/Meaning-making→Health behaviors	0.15 ^a^	0.07	0.13	0.23
Problem-focused coping→Meaning-making→Health behaviors	0.15 ^a^	0.04	0.08	0.24
Meaning-focused coping→Meaning-making→Health behaviors	0.13 ^a^	0.04	0.06	0.21

^a^ Empirical 95% confidence interval does not overlap with zero.

## Data Availability

The data presented in this study are available on request from the corresponding author.

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
