# Peer review of "Threat Appraisal, Resilience, and Health Behaviors in Recovered COVID-19 Patients: The Serial Mediation of Coping and Meaning-Making"

_ijerph, 2023, doi:10.3390/ijerph20043649_

Round 1

Reviewer 1 Report

This study investigates can coping strategies and meaning-making serially mediate the relationship of threat appraisal and resilience with health behaviors in recovered COVID-19 patients.

This manuscript is too long. There are more than 5500 words! I would suggest authors shorten the introduction and clarify the hypothesis. There are too many stated hypotheses (should be one, a maximum of two). Seven hypotheses make everything confusing and harder to read and understand. 

Introduction. The whole section is too long. Are all of those 72 references really necessary in the introduction? Five pages of introduction can't be crucial in describing all these behaviors and strategies. This is not a systematic review so authors should concentrate on only crucial sentences to present the topic and why their research is conducted and important. Figure 1 is almost the same as Figure 2 (I believe these two can be unified). 

Methods. The type of the study was not stated. Which serious medical conditions that would distort the person’s responses were really used as exclusion criteria? It should be clarified at this point. After how many days, weeks, or months after the COVID infection did patients fill out the questionnaire? Were those patients treated in hospitals or at home? What was the severity of their disease (it cannot be the same if a patient has mild symptoms or he/she needs oxygen therapy). Some basic information about where this research took place, when exactly, and how COVID was confirmed (rapid antigen test or PCR?) is missing. 

Results. In paragraph 3.2 there is no need to describe the meanings of all these correlations shown in table 2. I don't see any benefit from that. The first sentence in paragraph 3.3 belongs to paragraph 2.4 (Statistical analysis). Many sentences in the Results section actually describe why the statistical analysis was done that way and what was next, therefore results from this research are not presented as they should be. The "Results" section is no place for the author's comments, nor it should be used for detailed table and figure descriptions, or statistical analysis explanations. The whole "Results" section should be simplified, and rewritten in a more convenient way. 

Discussion. This section is very well written and should be the longest part of the research, unlike the introduction. There are a lot of limitations. It would be interesting seeing the differences between the patients after COVID infection and persons who did not have COVID but have a chronic disease.

Conclusion. The conclusion should be shortened, it is okay but too long. I don't think that I've ever seen a reference put in the conclusion section, especially at the end. This section should only conclude what this research discovered. Everything else should be in the discussion section. 

I like the general idea of this study but the manuscript must be seriously improved to be published in this journal. Good luck!

Author Response

Responses to Reviewer 1

This study investigates can coping strategies and meaning-making serially mediate the relationship of threat appraisal and resilience with health behaviors in recovered COVID-19 patients.

This manuscript is too long. There are more than 5500 words! I would suggest authors shorten the introduction and clarify the hypothesis. There are too many stated hypotheses (should be one, a maximum of two). Seven hypotheses make everything confusing and harder to read and understand. 

Introduction

1) The whole section is too long. Are all of those 72 references really necessary in the introduction? Five pages of introduction can't be crucial in describing all these behaviors and strategies. This is not a systematic review so authors should concentrate on only crucial sentences to present the topic and why their research is conducted and important. Figure 1 is almost the same as Figure 2 (I believe these two can be unified).

- The section of Introduction was shorten. We made 3 hypotheses, instead of 7 initially.

- We tried to concentrate only on crucial sentences. With respect to Figure 1 – it makes visually better understand the whole model, described in the last part of Introduction. We reached 53 references in Introduction instead of 72.

- In line with the suggestion regarding the figures, we deleted Figure 1 from our revision as most of the information is presented in Figure 2.

Methods 

2) The type of the study was not stated. Which serious medical conditions that would distort the person’s responses were really used as exclusion criteria? It should be clarified at this point. After how many days, weeks, or months after the COVID infection did patients fill out the questionnaire? Were those patients treated in hospitals or at home? What was the severity of their disease (it cannot be the same if a patient has mild symptoms or he/she needs oxygen therapy). Some basic information about where this research took place, when exactly, and how COVID was confirmed (rapid antigen test or PCR?) is missing. 

- In line with the first suggestion, we added in our revision that our study was cross-sectional in design. Next, while conducting our research we controlled the following four medical conditions that could distort the person’s responses and were used as exclusion criteria: cancer, cardiovascular disease, obstructive pulmonary diseases, and diabetes. They were chosen in accordance with the WHO recommendations (https://www.google.pl/url?sa=t&rct=j&q=&esrc=s&source=web&cd=&cad=rja&uact=8&ved=2ahUKEwifhbbDwo39AhXToVwKHQMkBuUQFnoECBkQAw&url=https%3A%2F%2Fwww.who.int%2Fnews-room%2Ffact-sheets%2Fdetail%2Fnoncommunicable-diseases&usg=AOvVaw33J15SJYEdDlJSqjwurHYf). The information was added in our revision.

- The time since COVID-19, after which patients completed questionnaires, ranged from 5 days to 18 months. It was corresponding to the time since COVID-19 in months that we gave in Table 1: the mean (M = 6,79) and standard deviation (SD = 4.29) of the time since COVID-19 is given in months. In response to the above-mentioned question we also added the range in our revision.

- The patients were treated both in hospitals or at home. In Table 1 we gave the number of patients who were hospitalised due to the COVID (N = 78); the rest of patients were treated at home (N = 188).

- In Table 1 we also included symptom severity, which was measured on a scale ranging from 1 to 10 (M = 5.78, SD = 2.34). The range was added in the revision.

- In line with the next suggestion, we added in our revision the information about where this research took place, when, and how COVID was confirmed.  We stated that our study was conducted from 1 November 2021 to 31 March 2022 in southern parts of Poland (Opole, Wroclaw, and Katowice), both privately by individuals in their homes and in outpatient clinics. COVID-19 was confirmed by two main types of viral tests: nucleic acid amplification tests (NAATs) and antigen tests.

Results

3) In paragraph 3.2 there is no need to describe the meanings of all these correlations shown in table 2. I don't see any benefit from that. The first sentence in paragraph 3.3 belongs to paragraph 2.4 (Statistical analysis). Many sentences in the Results section actually describe why the statistical analysis was done that way and what was next, therefore results from this research are not presented as they should be. The "Results" section is no place for the author's comments, nor it should be used for detailed table and figure descriptions, or statistical analysis explanations. The whole "Results" section should be simplified, and rewritten in a more convenient way. 

- In line with the suggestion, we moved the sentence “In the next step of statistical examination, we decided to test the serial mediation effects using SEM analysis.” from paragraph 3.3 to paragraph 2.4.

- We also simplified and rephrased the previously given information in the Results section in a more convenient way. In our revision we tried to avoid unnecessary interpretation of the results obtained. However, as our article was predominately psychological in nature, some of the information needed to comply with the main rules of reporting the results required by the APA 7th. For this reason, we tried to avoid our own interpretations, but only report the statistical test results obtained in accordance with the APA guidelines. Much of this seems necessary so that a reader who is not familiar with statistical principles can understand the results obtained. Otherwise, those without a statistical background would not be able to assimilate the results of our study. In addition, the other Reviewer suggested that we should add more information regarding the Result section, so we had to ensure a balance between both reviews.

As a whole, in the revision, the Result section is similar to other psychological articles using analogous statistical methods and tests.

Discussion

4) This section is very well written and should be the longest part of the research, unlike the introduction. There are a lot of limitations. It would be interesting seeing the differences between the patients after COVID infection and persons who did not have COVID but have a chronic disease.

Conclusion

5) The conclusion should be shortened, it is okay but too long. I don't think that I've ever seen a reference put in the conclusion section, especially at the end. This section should only conclude what this research discovered. Everything else should be in the discussion section. 

I like the general idea of this study but the manuscript must be seriously improved to be published in this journal. Good luck!

- In line with the suggestion, we removed unnecessary reference and abbreviated this section.

Reviewer 2 Report

The ms titled “Threat appraisal, resilience, and health behaviors in recovered COVID-19 patients: The serial mediation of coping and meaning-making” aimed to investigate the mediation role of coping strategies and meaning-making on the relationship between threat appraisal and resilience with health behaviors in recovered COVID-19 patients. Findings showed that problem-focused coping, meaning-focused coping, and meaning-making significantly mediated the relationship of threat appraisal and resilience with health behaviors. 

Congrats to the authors. I found this study very brilliant as well as the paper very well written. These findings may be a valuable source of knowledge for those researchers who deal with psychological wellbeing in recovered patients with COVID-19.  I feel the manuscript can be further improved and could be published after some minor changes. 

Introduction

Authors describe the role of coping strategies as predictors of well-being and mediator of the relationship between threat appraisal and healthy behaviours. I suggest deepening expand these sections by including evidence on the important role that coping strategies played on wellbeing and behaviours of healthy people (more in general) during the COVID-19 pandemic. Authors can consider those studies that directly assessed the above-mentioned effects (10.3390/ejihpe12090093; 10.1016/j.nepr.2020.102809; 10.12688/f1000research.73610.1) on participoants' well-being.

Statistical Analysis

Why the overall model has been adjusted for Age only while other relevant sociodemographic variables have been excluded (e.g., gender and education)?

Please, add the age variable to the correlation matrix. 

Did authors check for multicollinearity through VIF and Tolerance? If so, please, add this information.

Author Response

Responses to Reviewer 2

The ms titled “Threat appraisal, resilience, and health behaviors in recovered COVID-19 patients: The serial mediation of coping and meaning-making” aimed to investigate the mediation role of coping strategies and meaning-making on the relationship between threat appraisal and resilience with health behaviors in recovered COVID-19 patients. Findings showed that problem-focused coping, meaning-focused coping, and meaning-making significantly mediated the relationship of threat appraisal and resilience with health behaviors. 

Congrats to the authors. I found this study very brilliant as well as the paper very well written. These findings may be a valuable source of knowledge for those researchers who deal with psychological wellbeing in recovered patients with COVID-19.  I feel the manuscript can be further improved and could be published after some minor changes. 

Introduction

1) Authors describe the role of coping strategies as predictors of well-being and mediator of the relationship between threat appraisal and healthy behaviours. I suggest deepening expand these sections by including evidence on the important role that coping strategies played on wellbeing and behaviours of healthy people (more in general) during the COVID-19 pandemic. Authors can consider those studies that directly assessed the above-mentioned effects (10.3390/ejihpe12090093; 10.1016/j.nepr.2020.102809; 10.12688/f1000research.73610.1) on participants' well-being.

- Thank you for these suggestions. We included them in the part of Introduction, speaking about coping.

Statistical Analysis

2) Why the overall model has been adjusted for Age only while other relevant sociodemographic variables have been excluded (e.g., gender and education)?

̶  In line with the Reviewer’s suggestion, the overall model was adjusted for age, gender, and education. The path coefficients for problem-focused coping, meaning-focused coping, and meaning-making as serial mediators proved insignificant across age (χ2 = 1.01, p > .05), gender (χ2 = 1.23, p > .05), and education (χ2 = 0.94, p > .05). All this information was added in our revision,

3) Please, add the age variable to the correlation matrix. 

̶  We added the age variable to the correlation matrix in our revision . Thank you for this suggestion as the correlational result showed interesting findings, namely  that age was positively correlated with all variables except for threat appraisal.

4) Did authors check for multicollinearity through VIF and Tolerance? If so, please, add this information.

̶  Yes, as part of the preliminary analyses required for mediational analysis, we initially checked multicollinearity through VIF and Tolerance, but due to the multiplicity of reported data, we did not report these results in our original manuscript. In line with the Reviewer’s suggestion, we added the information regarding multicollinearity through VIF and Tolerance in our revision in the section of 2.4. Data analysis. The value of the variance inflation factor (VIF) collinearity test2 was 1.43, while the value of the tolerance coefficient was .67, which indicates the acceptable degree of correlation of the predictors entered for the analysis conducted.

Round 2

Reviewer 1 Report

The authors made a significant effort to improve this manuscript according to the reviewer's suggestions. All necessary changes have made it much easier to read and understand. Articles like this are infrequent but essential in understanding COVID-19 and its consequences, especially health behavior. The submitted manuscript in this form should be published. Congrats!